

# Incorporating reef fish avoidance behavior improves accuracy of species distribution models

Kostantinos A. Stamoulis[1], Jade M.S. Delevaux[2], Ivor D. Williams[3], Alan M. Friedlander[2,4], Jake Reichard[2], Keith Kamikawa[2] and Euan S. Harvey[1]

[1] Curtin University, Perth, Australia
[2] University of Hawaiʻi at Mānoa, Honolulu, HI, United States of America
[3] NOAA Pacific Islands Fisheries Science Center, Honolulu, HI, United States of America
[4] National Geographic Society, Washington D.C., United States of America

## ABSTRACT

Species distribution models (SDMs) are used to interpret and map fish distributions based on habitat variables and other drivers. Reef fish avoidance behavior has been shown to vary in the presence of divers and is primarily driven by spearfishing pressure. Diver avoidance behavior or fish wariness may spatially influence counts and other descriptive measures of fish assemblages. Because fish assemblage metrics are response variables for SDMs, measures of fish wariness may be useful as predictors in SDMs of fishes targeted by spearfishing. We used a diver operated stereo-video system to conduct fish surveys and record minimum approach distance (MAD) of targeted reef fishes inside and outside of two marine reserves on the island of Oʻahu in the main Hawaiian Islands. By comparing MAD between sites and management types we tested the assumption that it provides a proxy for fish wariness related to spearfishing pressure. We then compared the accuracy of SDMs which included MAD as a predictor with SDMs that did not. Individual measures of MAD differed between sites though not management types. When included as a predictor, MAD averaged at the transect level greatly improved the accuracy of SDMs of targeted fish biomass.

# INTRODUCTION

A current focus in marine ecology has been to use species distribution models (SDMs) to understand and sometimes predict fish distributions based on habitat, environmental, and other drivers. Development of SDMs begins with observations of species distributions (typically summarized in terms of biomass, biodiversity, or similar metrics), and the identification and compilation of environmental variables known or believed to influence habitat suitability and, therefore, species distributions (*Franklin, 2010*; *Schmiing et al., 2013*). Modeling approaches can be rule based or quantitative, and can include statistical models, multivariate ordination, classification, machine learning, and Bayesian techniques (*Norberg et al., 2019*). SDMs are often applied for predictive mapping, producing spatial

Corresponding author
Kostantinos A. Stamoulis,
kostanti@hawaii.edu

datasets and maps of species distributions and/or habitat suitability which, among other applications, can assist with marine spatial planning by identifying areas of high ecological significance (*Shucksmith & Kelly, 2014*; *Stamoulis & Delevaux, 2015*).

Fish species respond to their habitat in different ways depending on their life-history strategies, predators, competitors, and food availability (*Sale, 1998*; *Boström et al., 2011*). Fishing pressure is a primary driver, not only of fish distributions (*Jennings & Polunin, 1996*; *Friedlander & DeMartini, 2002*), but also (in the case of spearfishing) of fish behavior (*Kulbicki, 1998*). Fish behavior can be substantially altered by the presence of SCUBA divers, depending on fishes' prior experience of divers' activities (i.e., feeding vs spearing) (*Cole, 1994*; *Kulbicki, 1998*; *Watson & Harvey, 2007*). Consequently, it is reasonable to expect that such variability in fish behavior would influence survey counts from underwater visual census conducted by observers on SCUBA (*Brock, 1954*)—the most common survey method in shallow water coral reefs. Despite earlier recognition of the potential biases associated with variable responses of targeted fishes to divers (*Kulbicki, 1998*), there have only been a few attempts to quantify the impacts of fishes' diver avoidance behavior on measures of fish assemblages (*Dickens et al., 2011*; *Bozec et al., 2011*). Because fish assemblage metrics are response variables for SDMs, including measures of fish behavioral responses to the presence of survey divers may improve the predictive power of SDMs for targeted fishes.

Avoiding predators is a necessary and consistent behavior for reef fishes as well as most other animals (*Lima & Dill, 1990*). One of the most common avoidance behaviors is fleeing which can incur energetic and opportunity costs (*Ydenberg & Dill, 1986*). Human spear fishers are reef fish predators and targeted fishes avoid them in areas where spearfishing occurs, consistent with predation risk theory (*Gotanda, Turgeon & Kramer, 2009*). Predator avoidance behavior diverts time and energy away from health and fitness enhancing activities such as foraging, parental care, and mating (*Lima, 1998*; *Frid & Dill, 2002*) and can also have consequences for ecosystem functions (*Madin et al., 2010*; *Rizzari et al., 2014*). The application of a predation risk framework may help to interpret the extent to which underwater survey counts reflect actual abundances vs a behavioral bias (*Frid, McGreer & Frid, 2019*), the costs of avoidance behavior on body condition and reproduction (*Spitz et al., 2019*), and how these factors (together with fishery removals) may influence the results of SDMs.

In locations with high spearfishing pressure, area-based fish survey methods may underestimate abundance of species targeted by spear fishers (*Kulbicki, 1998*; *Feary et al., 2010*). Guam is a Pacific Island where SCUBA spearfishing has been practiced for over 30 years as part of the recreational and commercial reef-fish fisheries (*Myers, 1993*; *Houk et al., 2012*). *Lindfield et al. (2014)* tested the magnitude of avoidance behavior and its influence on fish surveys in Guam using a diver operated stereo video system (stereo-DOV) to survey fish populations inside and outside of two no-take reserves and compared counts using standard open-circuit SCUBA and a closed-circuit rebreather. Closed-circuit rebreathers do not produce bubbles when operated at a constant depth and, therefore, greatly reduce the disturbance caused by survey divers' presence and the association with SCUBA spear fishers. The authors recorded 'minimum approach distance' (MAD): the

distance between the diver and the fish at its closest point. In contrast to flight initiation distance, distance is recorded for every fish observed on a transect, even if the fish does not flee and moves away in what appears to be normal swimming activity (*Lindfield et al., 2014*). In addition, the observer moves at a steady pace along the transect and does not purposely approach fish with the goal of soliciting a flight response.

*Lindfield et al. (2014)* found that fished sites sampled on open-circuit SCUBA had the greatest average MAD for targeted fish groups and abundance of targeted fishes was 2.6 times greater when surveyed on closed-circuit rebreather vs. open-circuit SCUBA in fished areas, whereas in reserves, counts were similar between diving modes. This demonstrates a dramatic impact of fish behavior on survey estimates due to avoidance of open-circuit SCUBA diver 'predators'. These effects were partially corroborated by *Gray et al. (2016)* who used a different underwater visual census method and found that biomass estimates of some targeted reef fishes were significantly lower on open-circuit SCUBA compared to closed-circuit rebreather at locations with high spearfishing pressure in the main Hawaiian Islands.

Fishing has obvious and direct effects on targeted fish populations (*Jackson et al., 2001*). Patterns of fishing pressure are difficult to measure and are rarely mapped (but see *Stamoulis et al., 2018*). Diver avoidance behavior of targeted fishes may provide a proxy for spearfishing pressure (*Bergseth, Russ & Cinner, 2015*). Thus, inclusion of diver avoidance behavior in SDMs could have explanatory power beyond correcting underwater survey bias. Spearfishing pressure directly increases fish wariness and decreases *true* fish biomass, while increased fish wariness may further decrease *observed* fish biomass, due to survey diver avoidance. Therefore, including a measure of fish wariness should improve explanatory power and predictive accuracy of SDMs. In order to test this hypothesis, we used a stereo-DOV (*Goetze et al., 2019*) to (simultaneously) conduct belt-transects and record MAD of targeted reef fishes both inside and outside of two marine reserves on the island of Oʻahu in the main Hawaiian Islands. We compare MAD between reserve and fished areas and between sites to test the assumption that it provides a proxy for spearfishing pressure, then compare the accuracy of SDMs including MAD as a predictor with SDMs that do not.

## MATERIALS AND METHODS

### Study sites

Surveys were conducted inside and outside of two no-take marine reserves on Oʻahu in the Hawaiian Islands (Fig. 1). Pūpūkea is located on the north shore of Oʻahu and was originally established in 1983. It was 10 ha when first established and allowed for a range of fishing activities. In 2003 it was expanded to encompass 71 ha and fishing activities were prohibited. Spearfishing effort in the adjacent fished area to the north was estimated to be ~5,000 h/yr/km$^2$ (*Delaney et al., 2017*). Enforcement in this reserve is somewhat lacking and spearfishing has been documented inside the boundaries, though large seasonal ocean swells ensure there is little fishing during the winter months (*Stamoulis & Friedlander, 2013*). Surveys of Pūpūkea were conducted during June–October 2016. Hanauma Bay is

located on the south-east corner of the island and is the oldest MPA in the state, established in 1967. The entire bay is protected and encompasses 41 ha of marine habitats. Spearfishing effort in the adjacent fished area—Maunalua Bay—was estimated at ~250 h/yr/km$^2$ (*Delaney et al., 2017*) and the habitat is compromised due to urbanization and associated land-based impacts (*Wolanski, Martinez & Richmond, 2009*). The reserve is continuously monitored, and compliance is very high. Hanauma Bay was surveyed between February and May 2017. Both marine reserves are frequented by high numbers of recreational scuba divers and snorkelers although fish feeding is prohibited. Transect locations at both sites were randomly selected within management types (reserve and open) on hard-bottom habitats, spaced a minimum distance of 80 m apart, using ArcGIS (Fig. 1).

## Field surveys

Pre-determined survey locations were uploaded to GPS units for use in the field. Two divers navigated to waypoints from shore or small boat and used a stereo-DOV to conduct a single 5 × 25 m belt transect on SCUBA (Fig. 2). Each transect began on the selected GPS point and followed the depth contour. Transect length was measured using a 25 m line reel which was secured to the substrate at the beginning of the transect and rolled out as progress was made. Survey time was standardized to three minutes per transect. Field surveys were conducted under Hawai'i State special activity permit No. 2017-44.

Our stereo-DOV system used two Canon Legria HF G25 high-definition video cameras mounted 0.7 m apart on a base bar inwardly converged at 7° to provide a standardized field of view. These video cameras feature a 10 × HD video lens with a 30.4–304 mm (35 mm equivalent) focal length. Video was recorded at 1,920 × 1,080 (Full HD) resolution with a framerate of 25 frames/second. The camera system was built by and purchased from https://www.seagis.com.au. Stereo video imagery was calibrated using the program CAL (SeaGIS), following the procedures outlined in *Harvey & Shortis (1998)*. This allowed for measurements of fish length, distance (range), and angle of the fish from the center of the camera system, and standardization of the area surveyed (*Harvey, Fletcher & Shortis, 2001*; *Harvey et al., 2004*).

The stereo-DOV system recorded imagery while the observer moved along the transect, from which we measured the abundance, length, and MAD of all targeted reef fishes encountered on the transect. The observer held the stereo-DOV pointing forward and parallel with the bottom while swimming close to the substrate at a constant speed. Thus, each video-transect consisted of three minutes of (stereo) video captured while the observer moved along each 25 m transect. Fishes located greater than 10 m in front or 2.5 m to the left or right of the stereo-DOV system as it was moved along the transect were excluded based on minimum visibility encountered and transect dimensions (Fig. 2). Though visibility (water clarity) varied throughout the survey period, for consistency we applied the 10 m distance threshold to all surveys. We selected species targeted by spear fishers from the 'targeted' species classification of a recently published study of fishing effects in the main Hawaiian Islands, which included species with ≥450 kg of annual recreational or commercial harvest between 2000 and 2010, or that were otherwise recognized as important for recreational, subsistence, or cultural fishing (*Friedlander et al., 2018*, Table

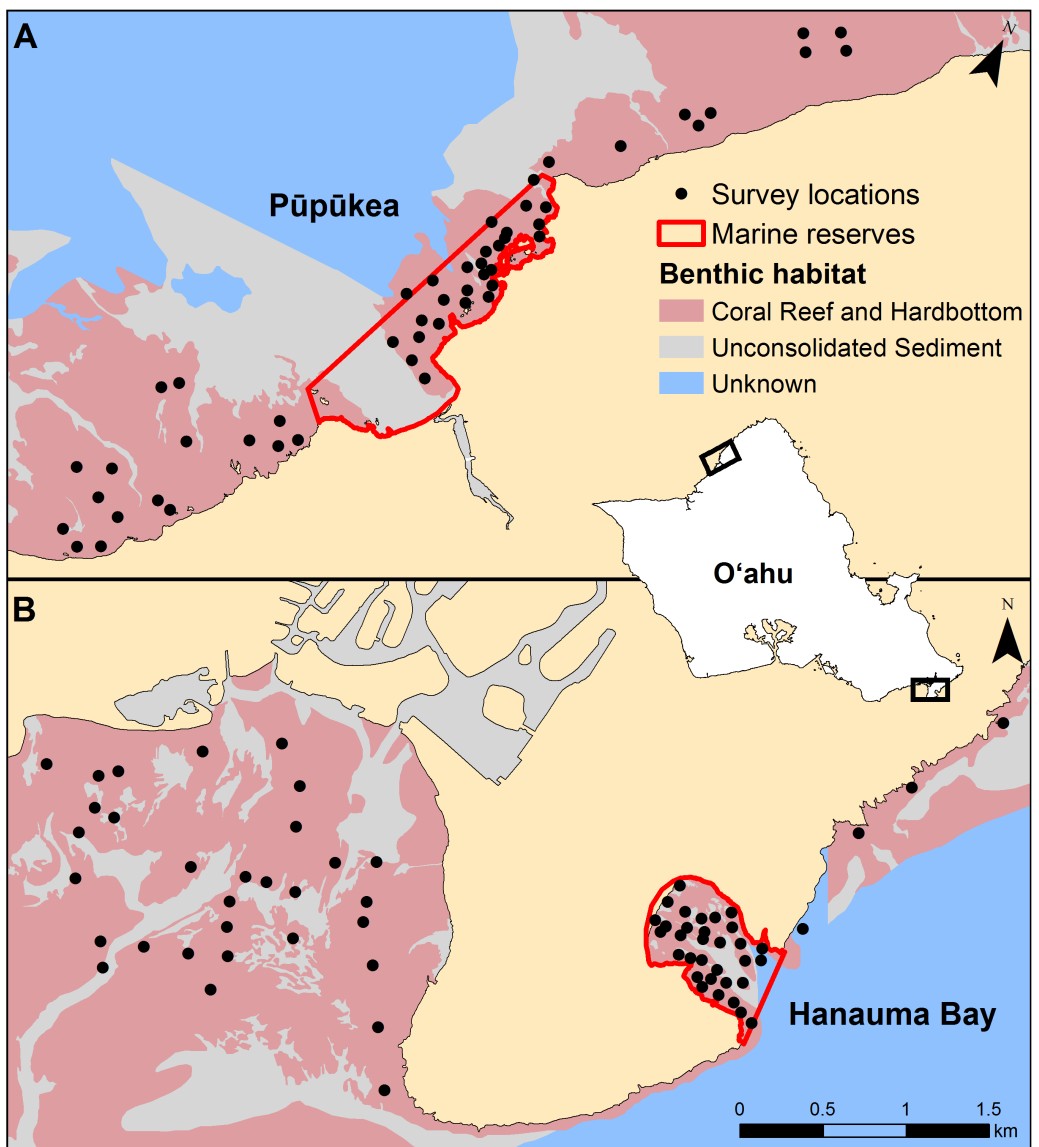

**Figure 1** **Transect locations at each study site.** (A) Pūpūkea and (B) Hanauma Bay. The inset map of the island of Oʻahu shows the map extents in black for each study site panel, which are the same scale.

S1). Full approval for this research was provided by the Curtin University Animal Ethics Committee in accordance with the Australian code for the care and use of animals for scientific purposes (Approval number: AEC_2014_42).

## Video analysis

Pairs of videos from the stereo-DOV system were analyzed using the program EventMeasure (SeaGIS). The total length of each targeted reef fish encountered on the transect was measured when the fish was closest to the stereo-DOV and computed by EventMeasure (*Harvey et al., 2004*). In the case of large schools, a representative subset of 6–10 individuals was measured, and the remaining fishes in the school were allocated to those records based

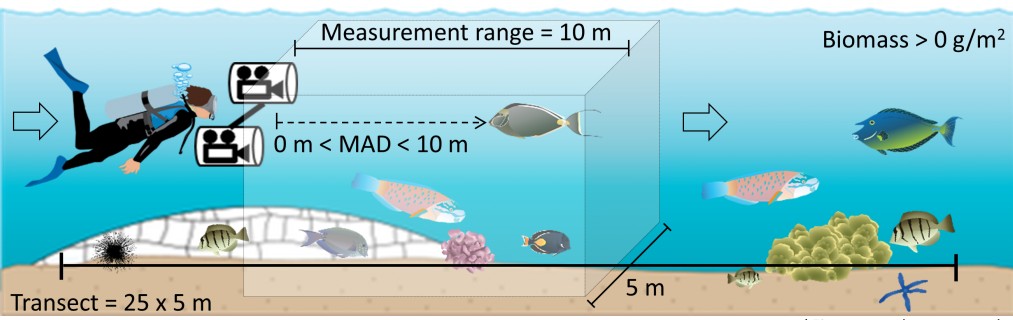

**Figure 2** **Diver operated stereo video (stereo-DOV) fish survey method.** Transect dimensions and measurement range of the stereo-DOV as the diver moves along the transect are shown. Species, number, length, and minimum approach distance (MAD) were recorded for each targeted fish species observed inside the transect dimensions and within the horizontal measurement range of the stereo-DOV. *Symbols courtesy of the Integration and Application Network, University of Maryland Center for Environmental Science* (ian.umces.edu/symbols/).

on size. Biomass was calculated from length estimates using the length-mass conversion: $M = aTL^b$, where parameters $a$ and $b$ are species-specific constants, TL is total length (cm), and M is mass (g). Length-mass fitting parameters were obtained from a comprehensive assessment of length-weight fitting parameters for Hawaiian reef fish species (*Froese & Pauly, 2017*). On transects where targeted species were not recorded, biomass estimates were set to zero.

### *Fish wariness (MAD)*
The shortest distance between the cameras and each targeted reef fish encountered on the transect was identified during the length measurement procedure (see above) and the distance was automatically computed by EventMeasure thus obtaining an accurate measurement of MAD (*Harvey et al., 2004*). If this was not possible due to the angle of the fish or obstruction of the camera view, another point was recorded and used to calculate MAD for the measured fish (*Lindfield et al., 2014*).

### Data analysis
### *Reserve effect*
To test effectiveness of the marine reserves included in this study as well as differences between sites, two-way ANOVAs were used to compare the effects of management (reserve vs fished) and site—the two study locations that each include a no-take reserve and an adjacent control area where spearfishing is permitted—on mean targeted fish biomass and body length by transect. To meet test assumptions, mean targeted fish biomass was fourth root transformed and mean targeted fish body length was ln $(x)$ transformed to improve normality. Morans I was used to test for spatial autocorrelation of mean biomass values between transects.

### Fish wariness (MAD)

A generalized linear mixed model was used to compare patterns of MAD between sites and management types and assess relationships with fish body length, angle of approach, and water depth. While management and site were the primary variables of interest for this study, it was necessary to include other variables which have been shown (or suspected) to influence fish flight behavior in Hawai'i (*Stamoulis et al., 2019*). These variables along with the interaction between management and site were included as fixed factors in the model, while transect number and species were included as random factors. Transect number was included to account for varying sample sizes by transect and species was included to account for species related differences in flight behavior. A gamma distribution with a log link function was applied as it was found to best fit the response variable distribution. Continuous variables were scaled prior to modeling and parameters were estimated with the Laplace approximation. To ensure that results of this model were not confounded by differences in targeted fish body length between sites and management types, a separate (otherwise identical) model was used to test these interactions.

### Species distribution models

SDMs were developed using $60 \times 60$ m resolution grid and all inputs were standardized accordingly. Survey variables were summarized at the transect level and these values were attributed to the corresponding grid cell. Targeted reef fish biomass was summed, and MAD was averaged for each transect. Boosted regression trees were used to develop SDMs of the total biomass of targeted reef fish for both study areas combined. SDMs and spatial predictions were generated in R (*R Core Team, 2014*) using the dismo (*Hijmans et al., 2014*) and raster (*Hijmans, 2014*) packages. Boosted regression trees are effective at modeling nonlinearities, discontinuities (threshold effects) and interactions between variables (*Breiman, 1996*; *Breiman, 2001*; *De'ath & Fabricius, 2000*). Targeted reef fish biomass was modeled using a Gaussian distribution and a fourth root transformation was applied to improve normality.

Model fitting and selection was accomplished following the procedures detailed in *Elith, Leathwick & Hastie (2008)*. To increase parsimony, selected models were then simplified to remove less informative predictor variables (*Elith, Leathwick & Hastie, 2008*). Starting with the selected best model and full set of predictor variables, the predictor contributing the least was identified and dropped, the model was re-fit, and the change in predictive deviance was calculated relative to the initial model. This process determined how many predictors could be dropped without resulting in a reduction of predictive performance. Simplification generally resulted in models with <10 predictors. Models with a larger number of predictors tend to have higher percent deviance explained. To allow for comparison, the top eight predictors were retained for all models. Then, the model training dataset was repeatedly sampled with replacement to create 20 bootstrap samples. Using the optimal parameter value combination and simplified set of eight predictor variables, a boosted regression tree model was fit to each bootstrap sample and used to make predictions based on the values of the predictor variables at each transect location. The mean of the bootstrapped predictions was used for interpretation and further analysis.

**Table 1** **Habitat predictors used in species distribution models (SDMs).** Number of individual datasets of each type indicated in parentheses. See Table S2 for additional details.

| Predictor dataset types | Datasets | Description |
| --- | --- | --- |
| Seafloor topography (12) | Depth, Slope, Slope of slope, Aspect, Planar and Profile curvature, Bathymetric position index | Seafloor topography metrics derived from bathymetry including depth, slope, structural complexity, exposure, curvature, and bathymetric position index. Slope, slope of slope, and bathymetric position index were calculated at 60 m and 240 m. |
| Benthic habitat composition (7) | Percent cover of Coral, Crustose coralline algae, Macroalgae, Turf, and Soft bottom, Proximity index, Shannon's diversity index | Percent benthic cover of major cover types, seascape fragmentation/patch isolation, habitat diversity. |
| Geographic (3) | Latitude, Longitude, Distance to shore | Geographic location and distance from shore. |
| Oceanographic (1) | Wave Power | Wave height × wave period. |

Habitat variables were those used in (*Stamoulis et al., 2018*) following a pairwise correlation analysis for the Main Hawaiian Islands and conversion to a 60 × 60 m resolution grid. There were 23 total habitat variables of four broad categories: seafloor topography, benthic habitat composition, geographic, and wave energy (Table 1, see Table S2 for further details). Four transects in the open area near Hanauma Bay did not have remotely sensed habitat data and were excluded from SDMs.

To determine whether including behavior as a predictor improved model fit and predictive performance, models were developed separately using predictor sets that included and excluded MAD. In addition to the habitat variables described above, management type (reserve/open) was included as a predictor after testing its correlation with MAD. In summary, two boosted regression tree models were developed combining data from both sites to explain and predict targeted fish biomass; (1) habitat + management, and (2) habitat + management + MAD.

Model fit was evaluated using cross-validated percent deviance explained and cross-validated standard error. Predictive performance was assessed by comparing predicted values to observed values for each location using $R^2$ and Gaussian rank correlation estimate (*Boudt, Cornelissen & Croux, 2012*), as well as root mean square error and symmetric mean absolute percent error-an alternative to mean absolute percent error that is robust to zero values.

## RESULTS

### Sampling and reserve effect

Stereo-DOV belt transect surveys were conducted inside the marine reserves and in the adjacent open areas at both Pūpūkea and Hanauma Bay (Table 2). These resulted in a total of 1,486 observations of 35 coral reef fish species targeted by spear fishers in Hawaiʻi (Table S1). Reserve locations had higher abundances of targeted species such that the majority of observations occurred at locations protected from fishing (Table 2). At Hanauma Bay, 25% of transects had no targeted fishes and at Pūpūkea 13% of transects had no targeted fishes. With few exceptions, these transects were located in the fished areas at each study

**Table 2  Transect and sample numbers of targeted fishes by site and management type.**

| Site | Management | Transects | Fishes recorded |
|------|-----------|-----------|-----------------|
| Pūpūkea | Reserve | 25 | 475 |
| | Open | 27 | 272 |
| Hanauma Bay | Reserve | 35 | 572 |
| | Open | 37 | 167 |
| | *Total:* | *124* | *1,486* |

**Table 3  Generalized linear mixed model results for minimum approach distance (MAD) combining both sites.** Fixed effects include management type, site, fish body length, transect depth, approach angle, and the interaction between management and site (Mgmt × Site). Columns represent the fixed effect estimate (Estimate), standard error (Std. error), $t$-value, and $p$-value.

| | Estimate | Std. error | $t$-value | $p$-value | |
|------|----------|-----------|-----------|-----------|---|
| Management | −0.08 | 0.12 | −0.7 | 0.491 | |
| Site | −0.28 | 0.13 | −2.2 | 0.025 | * |
| Fish length | 0.21 | 0.04 | 5.4 | <0.001 | *** |
| Depth | 0.17 | 0.08 | 2.3 | 0.024 | * |
| Angle | 0.13 | 0.03 | 5.0 | <0.001 | *** |
| Mgmt × Site | −0.07 | 0.16 | −0.5 | 0.640 | |

site. Marine reserves had significantly higher biomass of targeted fishes ($F_{1,120} = 48.9$, $p < 0.001$) compared to adjacent fished areas, though the magnitude differed. The ratio of mean targeted fish biomass inside the reserve vs. outside was 4.9 for Hanauma Bay and 1.5 for Pūpūkea. In contrast, there was no significant difference in mean targeted fish body length for marine reserves compared to adjacent fished areas ($F_{1,95} = 1.0, p > 0.05$). There was also no difference between sites for mean biomass ($F_{1,120} = 2.1, p > 0.05$) or mean body length ($F_{1,95} = 3.2, p > 0.05$). Morans I test for spatial autocorrelation indicated spatial independence of measured biomass values ($Z = 1.1, p > 0.05$).

### Fish wariness (MAD)

MAD ranged from 0.8 to 10 m and was not significantly different inside and outside of reserves, though differed between sites (Table 3). MAD at Pūpūkea was significantly higher overall compared to Hanauma Bay (Fig. 3D). Fish length, depth, and angle of approach were all significantly positively related to MAD (Table 3, Fig. 3). Fish length ranged from 4 to 70 cm, transect depth ranged from 0.5 to 17 m, and angle of approach ranged from 0 to 25 degrees. The model that included interaction terms of targeted fish body length with site and management type, respectively, showed no significant effect of either Fish length × Site ($t = 1.4, p > 0.05$) or Fish length × Mgmt ($t = −1.9, p > 0.05$).

### Species distribution models

Management type was not correlated with MAD (Spearman rho $= 0.1$). The model that included management, but not behavior explained 31% of the variability in targeted fish biomass for Hanauma Bay and Pūpūkea combined (CV PDE, Table 4). For the model where MAD was included as a predictor, CV PDE increased by 38% (Table 4). For this

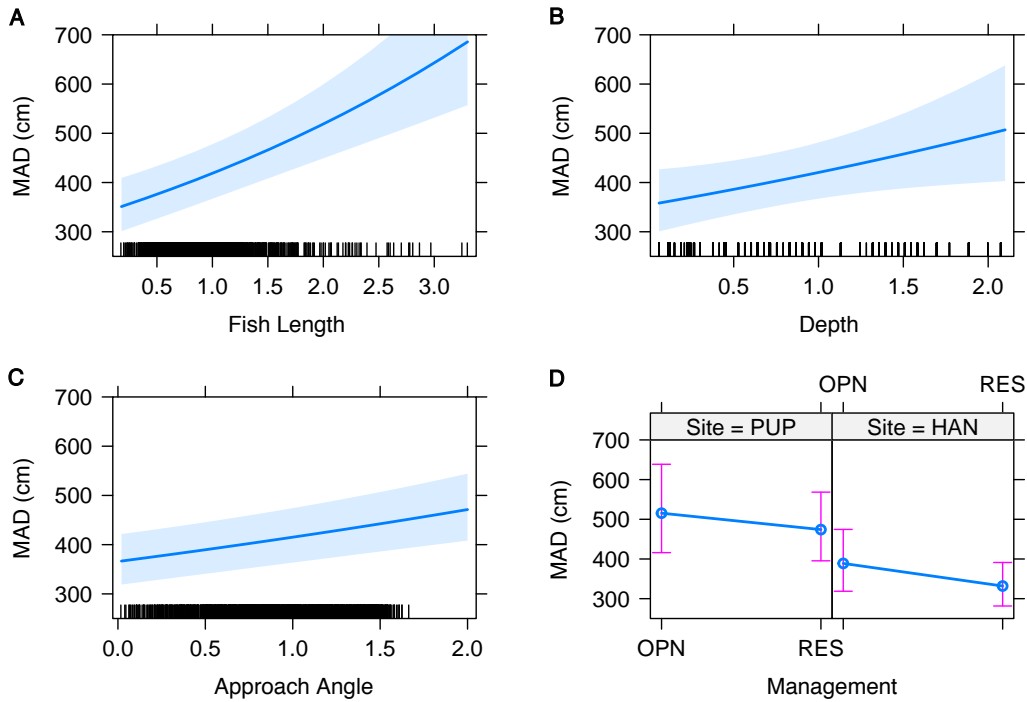

**Figure 3  Fixed effects of the generalized linear mixed model of minimum approach distance (MAD) for both sites combined.** Fixed effects and measured ranges include (A) fish length (4–70 cm), (B) transect depth (0.5–17 m), (C) angle of approach (0–25 degrees), and (D) management type (OPN = fished area, RES = marine reserve) by site (PUP, Pūpūkea; HAN, Hanauma Bay). A 95-percent confidence interval is drawn around the estimated effect. All continuous variables were scaled prior to modeling and the small lines along the *x*-axis show the distribution of data for each variable.

model, MAD accounted for 71% of explained variation for both sites combined (Fig. 4) and prediction accuracy increased with larger values of $R^2$ and GRCE compared to models which did not include MAD (Table 4). Prediction error for all three measures decreased when MAD was added to the model (Table 4) and MAD explained the greatest amount of variability compared to other predictors (Fig. 4). In the model management status but not behavior, management was not selected as a final predictor.

# DISCUSSION

## Management and site differences in fish wariness

Measurements of MAD did not differ significantly by management type, though they did differ between sites, with larger values at the site with higher spearfishing pressure. These results are consistent with the hypothesis that MAD is a proxy of fish wariness that increases with spearfishing pressure and correspond to those of *Lindfield et al. (2014)* who compared MAD of targeted fishes between reserves and fished areas in Guam, and *Goetze et al. (2017)* who measured MAD of targeted species before and after harvest events in periodically harvested closures in Fiji. In the latter study, increases in fish wariness were evident across all size ranges of targeted species when fish drives—where villagers work together to drive

**Table 4** **Species distribution model (SDM) evaluation comparison for models including management (Mgmt) and management and behavior (minimum approach distance—MAD).** Accuracy metrics include cross validated percent deviance explained (CV PDE), adjusted $r$-squared ($R^2$), and gaussian rank correlation estimate (GRCE). Error metrics include cross-validated standard error (CV SE), root mean square error (RMSE) and symmetric mean absolute percent error (SMAPE).

|  | Mgmt | MAD |
|---|---|---|
| **Accuracy** | | |
| CV PDE | 30.5 | 68.5 |
| $R^2$ | 0.37 | 0.74 |
| GRCE | 0.79 | 0.91 |
| **Error** | | |
| CV SE | 10.7 | 5.4 |
| RMSE | 32.2 | 25.2 |
| SMAPE | 1.04 | 0.91 |

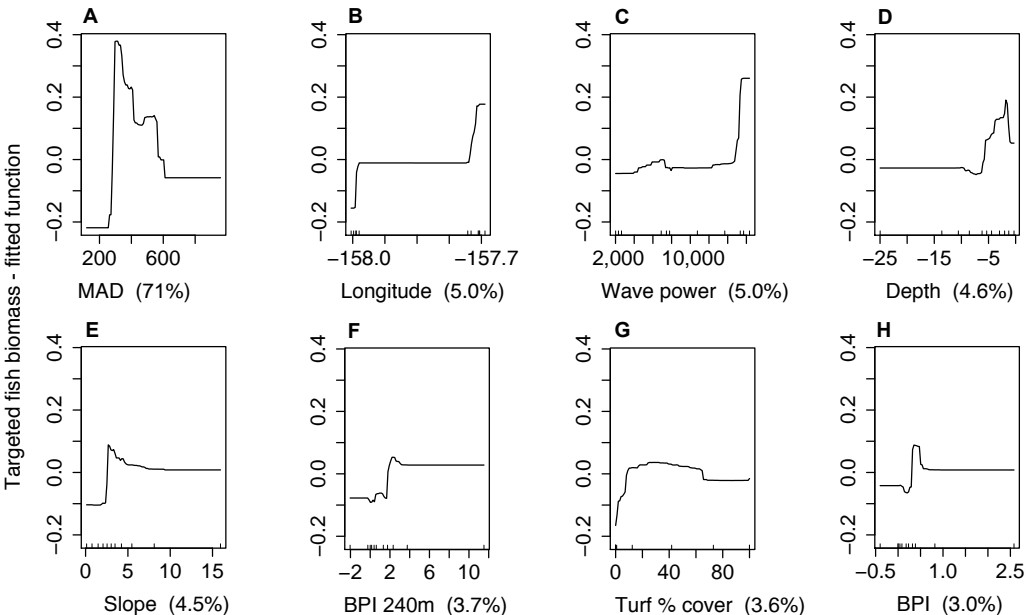

**Figure 4** **Partial dependence plots for the boosted regression tree model of targeted fish biomass including minimum approach distance (MAD) for both sites combined.** Predictor variables and units include (A) minimum approach distance [cm], (B) longitude [decimal degrees], (C) wave power [Kw/m], (D) depth [m], (E) slope [degrees], (F) bathymetric position index—240 m radius [m], (G) turf algae percent cover [%], and (H) bathymetric position index—60 m radius [m]. The y-axis represents the fitted function for targeted fish biomass. Panel labels identify predictor variables with percent variability explained in parenthesis. Small lines along $x$-axis show the distribution of each predictor.

fish into gillnets—were used as a harvest method. In contrast, spearfishing primarily affected larger individuals (*Goetze et al., 2017*). Fish drives are rarely utilized in modern times in Hawai'i and not at the sites included in this study, so we assume spearfishing to be the primary cause of increased fish wariness to human divers and do not expect an influence from passive capture methods such as hook and line.

MAD was significantly higher on average at Pūpūkea on the north shore of Oʻahu, compared to Hanauma Bay on the south shore. A likely explanation is that spearfishing pressure is also higher at Pūpūkea, both outside and inside the reserve. *Januchowski-Hartley et al. (2015)* showed that FID increased with fishing pressure in both fished areas and adjacent marine reserves in the Indo-Pacific, providing evidence of behavioral spillover which may also help to explain this pattern. Surveys at Pūpūkea were conducted in the summer months when the wave conditions allow for diving/spearfishing and the shoreline at the Pūpūkea reserve is very accessible with multiple access points. Spear fishers can swim in from either boundary or simply enter the reserve directly, and illegal spearfishing has been documented (*Stamoulis & Friedlander, 2013*). In contrast, shoreline access to the Hanauma Bay reserve is highly regulated and it is unlikely that any illegal spearfishing occurs, with the possible exception of divers crossing the seaward boundary from boats. Furthermore, spearfishing pressure in the area adjacent to the Hanauma Bay reserve is estimated to be only 5% of spearfishing pressure in the area adjacent to the Pūpūkea reserve. Thus, low compliance at Pūpūkea reserve and low spearfishing pressure in the area adjacent to Hanauma Bay are likely responsible for the larger effect of site than management on MAD in this study. Likewise, low compliance at Pūpūkea likely contributes to the small relative difference in targeted fish biomass between the reserve and open areas compared to Hanauma Bay where low biomass in the open area is presumably due to poor habitat as much as fishing.

## Effects of other variables on fish wariness

Fish body length had a positive relationship with MAD as shown in previous studies (*Lindfield et al., 2014*; *Goetze et al., 2017*). Reproductive value often increases with size in fishes (*Birkeland & Dayton, 2005*), and theory predicts that risk-taking should decrease at higher levels of reproductive value (*Clark, 1994*). In addition, larger fishes are often preferentially targeted by fishers and may have more experience with this threat, so are more willing to incur fleeing costs compared to smaller fishes (*Tsikliras & Polymeros, 2014*; *Samia et al., 2019*). Previous studies using flight initiation distance (FID) as a measure of fish wariness also showed a positive relationship with body length (*Gotanda, Turgeon & Kramer, 2009*; *Januchowski-Hartley et al., 2011*; *Januchowski-Hartley et al., 2015*; *Bergseth et al., 2016*). The strong positive relationship between fish body length and MAD could lead to concerns that modeled differences in MAD between sites and management types may be confounded. While including fish body length as a fixed factor in the generalized linear mixed model should have accounted for body length effects, a separate model showing no significant effects of the interactions between fish body length with site and management confirmed that modeled differences were based on MAD and not fish body length. In addition, comparisons of mean targeted fish body length by transect showed no differences between sites or management types.

Approach angle ranged from 0 to 25° and had a significant positive relationship with MAD. Fishes measured at a more oblique (higher) angle are farther from the transect and are consequently less likely to be approached closely compared to fishes nearer to the transect. In addition, the predation risk framework predicts a greater FID when approaches

are more direct because a direct approach may indicate detection and intent to capture (*Frid & Dill, 2002*). It follows that MAD is also influenced by approach angle in this way since both are measures of fish wariness. In contrast to our results, *Goetze et al. (2017)* did not find a relationship between MAD and approach angle.

Depth had a positive relationship with MAD. This is contrary to previous findings which showed depth to have a negative relationship with FID (*Stamoulis et al., 2019*). This effect is likely context dependent, and the positive influence of depth in this study reflects the low MAD in shallow areas of the marine reserves surveyed in this study. Both Hanauma Bay and Pūpūkea receive a large number of visitors who come to enjoy the abundant marine life. The majority of tourists tend to remain in shallow areas, thus targeted fishes in these marine reserves are likely habituated to non-aggressive human interactions, leading to reduced MAD in shallow areas. In contrast, the marine reserve surveyed by *Stamoulis et al. (2019)* has restricted access and does not receive many visitors. These findings suggest that fish flight behavior can be mediated by human interactions even in the absence of spearfishing (*Frid & Dill, 2002*; *Titus, Daly & Exton, 2015*).

## MAD as predictor for species distribution models

Including MAD as a predictor for SDMs greatly improved model fits and predictive performance. Despite some outliers at the low end of the scale, the partial dependence plot indicated an overall negative relationship between MAD and targeted fish biomass. In contrast, management type was not selected as a final predictor for either model. Though we showed a significant effect of management on targeted fish biomass, this suggests that habitat variation within management types is an important driver. Likewise, mean MAD explains variation among individual transects ($N = 120$) while management status explains variation only between management types ($N = 2$) and binary variables tend to have low explanatory power. We know that compliance differs between marine reserves in this study, so a dichotomous management status designation may be somewhat misleading. MAD was greater at the site with higher spearfishing pressure and had a negative relationship with targeted fish biomass when included in SDMs. Based on these results, mean MAD of targeted species at the transect level appears to be a robust measure of fish wariness when used in SDMs of targeted fish biomass.

The predation risk framework suggests that lower targeted fish biomass in fished areas may be due to the combined effect of fishery removals and the costs of antipredator behavior, in addition to potential survey bias due to avoidance behavior. Antipredator behaviors have the benefit of increasing survival in the face of predation risk, and the cost of diverting time and energy from foraging or other fitness enhancing activities (*Lima & Dill, 1990*; *Clark, 1994*). Our results indicate that MAD provides a measure of avoidance behavior that increases with perceived risk (spearfishing pressure), consistent with the economics of flight distance (*Ydenberg & Dill, 1986*). With additional data and species-specific analyses, application of the predation risk framework may help to generate bias correction factors accounting for fish behaviors that can be related to individual and species characteristics that influence investment in antipredator behavior (*Frid, McGreer &*

*Frid, 2019*). For use in SDMs, it appears that including MAD as a predictor helps to account for behavioral survey bias and improve model accuracy and predictive performance.

It is unclear what portion of the variance explained by MAD in SDMs was due to survey bias from fish behavior and what portion was due to the direct effects of spearfishing pressure, for which MAD provides a proxy. However, because the direction of these influences on observed targeted fish biomass are the same (negative), it is irrelevant to SDM performance. In order to validate the use of MAD as a proxy for spearfishing, future research should focus on comparing empirical measures of spearfishing pressure with MAD of targeted species to better quantify this relationship. A drawback of using MAD as a predictor for SDMs is that it is not possible to make predictions to locations for which MAD data is not available. Instead, spatially explicit estimates of spearfishing pressure could be used directly as a predictor for SDMs (e.g., *Stamoulis et al., 2018*). A better understanding of the relationship of MAD and spearfishing pressure would help inform this work so that MAD could be used to ground-truth spatial models of fishing pressure.

Another possibility is integrating MAD directly into measures of fish assemblage characteristics used to calibrate SDMs. Distance-based sampling, which is widely used for terrestrial mammals and birds but less so for coral reef fishes (though see *Kulbicki, 1998*; *Kulbicki et al., 2010*), is one approach that may allow incorporation of MAD. Specifically, in distance sampling, observers record the distance of each organism of interest from the observer at the time of observation, thereby incorporating an indirect measure of behavior (*Buckland et al., 2005*). Creating a detection function, representing the probability of detection as a function of distance from the line, allows for estimation of the proportion of fish missed within the surveyed area, resulting in corrected density estimates (*Buckland et al., 2005*). In this case, detection functions could be generated using data from locations with no spearfishing pressure, which should correct for altered fish behavior when applied in areas where spearfishing occurs, thus generating more accurate density estimates for use in SDMs.

Alternatively, the use of miniature remotely operated vehicles (mini-ROVs) to sample fish populations may address many of the effects of diver avoidance behavior. While mini-ROVs will still move and create a visual stimulus that may create trade-offs similar to those associated with predation risk (*Frid & Dill, 2002*), they do not produce bubbles and are <1 m in length, (*Sward, Monk & Barrett, 2019*), thereby removing the threat and much of the disturbance stimuli associated with human divers. Mini-ROVs can be outfitted with stereo-video systems for fish surveys and measurement of MAD (eg., *Schramm et al., 2020*). *Raoult et al. (2020)* compared underwater visual census results from mini-ROVs and human snorkelers and found that mini-ROV surveys detected greater abundance and diversity of fishes. Further research should compare stereo-video fish surveys conducted by mini-ROV to those conducted by human divers, such as in this study. MAD in particular should be compared to determine differences in fish flight behavior between methods and ascertain to what extent mini-ROVs can reduce survey bias associated with human divers and produce more accurate data for use in SDMs and other applications.

## CONCLUSIONS

In this study, we tested whether using a measure of targeted fish wariness (MAD) as a predictor of targeted fish biomass in SDMs spanning marine reserve boundaries, improved explanatory power and predictive accuracy. Our results show that including mean MAD as a predictor in SDMs greatly improves model performance and accuracy compared to models using reserve status only. Diver operated stereo-video systems allow for efficient sampling of reef-fish assemblages as well as fish behavior and do not require extensive training, making them useful monitoring tools for managers and communities. Based on the results from this and two previous studies (*Lindfield et al., 2014*; *Goetze et al., 2017*), MAD appears to be a useful proxy for spearfishing pressure. In order to fully validate MAD as a proxy for spearfishing, future research should focus on comparing empirical measures of spearfishing effort with MAD of targeted species. In addition, research should seek to improve spatially explicit estimates of spearfishing pressure, for which MAD could provide a valuable reference, to extend SDM predictions to un-sampled areas.

## ACKNOWLEDGEMENTS

Many thanks to those that assisted with fieldwork: Andrew Purves, Jonatha Giddens, Whitney Goodell, Jackie Troller, and Ignacio Petit. We would like to thank Catherine Gewecke (HI Division of Aquatic Resources) for facilitating the state permitting process and to Kaipo Perez (Hanauma Bay) for his help and coordination. We'd also like to thank Dr. Mark Hixon for sharing his mooring in Hanauma Bay and Kara Miller for lending the use of her kayak to survey the shallow sites in Maunalua Bay.

### Funding

This work was supported by an Australian Government Research Training Program Scholarship through Curtin University. The funders had no role in study design, data collection and analysis, decision to publish, or preparation of the manuscript.

### Grant Disclosures

The following grant information was disclosed by the authors:
Australian Government Research Training Program Scholarship through Curtin University.

### Competing Interests

The authors declare there are no competing interests.

### Author Contributions

- Kostantinos A. Stamoulis conceived and designed the experiments, performed the experiments, analyzed the data, prepared figures and/or tables, authored or reviewed drafts of the paper, and approved the final draft.
- Jade M.S. Delevaux performed the experiments, prepared figures and/or tables, authored or reviewed drafts of the paper, and approved the final draft.

- Ivor D. Williams and Euan S. Harvey conceived and designed the experiments, authored or reviewed drafts of the paper, and approved the final draft.
- Alan M. Friedlander conceived and designed the experiments, performed the experiments, authored or reviewed drafts of the paper, and approved the final draft.
- Jake Reichard performed the experiments, analyzed the data, authored or reviewed drafts of the paper, and approved the final draft.
- Keith Kamikawa performed the experiments, authored or reviewed drafts of the paper, and approved the final draft.

### Animal Ethics

The following information was supplied relating to ethical approvals (i.e., approving body and any reference numbers):

The Curtin Animal Ethics Committee provided full approval for this research in accordance with the Australian Code for the Care and Use of Animals for Scientific Purposes (approval number: AEC_2014_42).

### Field Study Permissions

The following information was supplied relating to field study approvals (i.e., approving body and any reference numbers):

Field surveys were conducted under Hawaii State special activity permit No. 2017-44.

### Data Availability

The raw measurements are available in Supplemental Files.

### Supplemental Information

Supplemental information for this article can be found online at http://dx.doi.org/10.7717/peerj.9246#supplemental-information.

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
