# Peer review of "Incorporating reef fish avoidance behavior improves accuracy of species distribution models"

_PeerJ, doi:10.7717/peerj.9246_

## Round 0.1 · original submission · Major Revisions

Overview

Although PeerJ normally considers two reviews sufficient, we obtained three reviews of your manuscript as a result of nearly simultaneous acceptance of invitations by the second and third reviewers. The reviews overlap on several major points but each also has many helpful suggestions not mentioned by the others. I have some additional comments largely related to the clarity of the presentation.

Editor’s Comments

Abbreviations: I agree with the reviewer that there are too many acronyms. This impedes the ease of reading. Please remove any that are not essential. For example, UVC is used only twice, CCR three times, MHI and DAR only once. MAD, stereo-DOV, SDM, are used often enough to be acceptable as abbreviations.

The figures and tables and their captions/headings need some work.
Fig. 1. Use capital letters for panel identification following PeerJ instructions to authors. Move the letters inside the panel frames to reduce wasted space and thus allow figures to be larger on the page. (You will probably have to move the compass symbols.) The caption should direct attention to the inset map of the island and the black rectangles within it.
Fig. 2. Because figures should be comprehensible without reference to the text, define stereo-DOV and MAD in the caption.
Fig. 3. Caption is incomplete. Define LMM and MAD. Define the continuous line, gray around the line, small lines along the x-axis, lines showing variation in lower right panel (which don’t need to be red), and clarify what angle and depth the x-axes refer to. Was fish length scaled and centered even though it does not seem to have many, if any, values below zero? Add capital letters to the panels and refer to each panel in the caption. Move the panels closer together vertically and horizontally, avoiding excessive crowding; unnecessary white areas simply force figures to be smaller on the page. It appears that you are using some sort of transformation of the y-axis as the lines do not appear to be equidistant; please make it explicit in the caption.
Figs. 4-6. Improve the captions following suggestions for preceding figures.
Table 1. Define abbreviations and acronyms. I think the diversity index is usually called the Shannon Index. If you prefer Shannon’s, insert the apostrophe. I think this is the only place where Proximity Index is mentioned. Define it in methods (or a footnote here if appropriate). Not clear what the reader gains from the reference to measures at two scales. What are the scales and how were they used here? Since these are your predictors, I think that the units are needed.
Tables 3 and 4. Define all acronyms and abbreviations. I think that P-value and F-value require hyphens.

I have highlighted a few references in which capital letters are retained after the first word in article titles.

I have included an annotated pdf which indicates several other minor issues of grammar and style.

Reviewer 1 ·

Basic reporting

I find the writing to be unclear in various key sections (including methods). I elaborate on this in my comments to the author.

The referenced literature does not adequately address the large body of work in which a predation risk framework is used to predict and interpret behavioral responses to human disturbance.

The article has an adequate structure and raw data are shared.

The manuscript is self-contained and an appropriate unit of publication.

Experimental design

The manuscript represents original research within the aims and scope of the journal.

Research questions are well defined and relevant. The text identifies the knowledge gap being addressed.

While the investigation is generally rigorous, I do raise issues with some of the methodology used (see comments to author).

Not all aspects of methods were described adequately. (see comments to author)

Validity of the findings

While the study is potentially important for evidence-based marine conservation, I do raise questions about the validity of some of the findings (see comments to author).

Data are provided and their collection appears to be sound (with some clarifications required). I do question, however, the validity of some of the analyses (see comments to author)

Conclusions vary in the extent to which I agree with them in terms of support from the results (see comments to author)

Speculation is not unreasonable.

Additional comments

This manuscript reflects a lot of intricate data collection occurring both in the field and while processing stereo camera images, and I commend the authors for it. The study’s objectives can be summarized as 1) testing whether the minimum approach distance (MAD) between divers and individual fish is higher where fish are exploited by divers with spearfishing gear than in protected areas, and 2) whether the explanatory power of species distribution models increases if MAD is used as a predictor. The first objective is corroborative of earlier studies, while the second is more novel.

While the manuscript has the potential to contribute to evidence-based marine conservation, its current form is insufficient to do so for the following reasons. First, the manuscript is written in a vacuum of the predation risk framework used to predict and interpret the behavior of fish or other animals in response to disturbance by humans (e.g. (Frid & Dill 2002; Samia et al. 2019; Spitz et al. 2019). The application of this framework may help interpret the extent to which species distribution models reflect actual abundances vs bias in survey counts, and potentially lead to correction factors (see, for instance, Frid et al (2019)). Additionally, a predation risk framework could guide discussion on the extent to which spatial variation in MAD may reflect the costs of antipredator behaviour on body condition and reproduction (Heithaus et al. 2008; e.g. Spitz et al. 2019), and how these factors (in conjunction with fishery removals) may influence the results of species distribution models.

I am also concerned about aspects of the analyses which were explained poorly or which are potentially inadequate. Additionally, the writing is unclear or insufficient on various conceptual and methodological aspects essential to the study. Below I elaborate on these points.

Introduction.
In general, the context of predation risk and its applications to the study’s objectives should be added (see my earlier comments).

L38-40 state that “Fishing pressure is a primary driver, not only of fish distributions (Jennings and Polunin 1996, Friedlander and DeMartini 2002), but also of fish behavior (Kulbicki 1998).” As implied by the referenced literature, this statement applies to tropical reefs where spearfishing is the dominant exploitation method. In most other parts of the world, however, most fishing occurs via deployment from vessels of other gear types (nets, hooks). Thus, all aspects of the paper (including title and abstract) should be re-written more transparently about the more limited scope of the paper (i.e., explicit to areas where spearfishing by divers is the dominant exploitation method).

L52-64: while the rebreather vs non-rebreather example is interesting the inherent mechanism on MAD differs from that of risk from spear fishers, and this needs to be contextualized.

L57-58: The definition of ‘minimum approach distance’ (MAD) is essential, yet here it is introduced merely as “the distance between the diver and the fish at its closest point”, which is ambiguous and potentially problematic. The definition should specify that MAD is the distance from the observer at which the individual fish engage on avoidance behavior (i.e. flee). This definition makes it explicit when fish chose to incur the costs of antipredator behavior. If the authors expect fleeing to not occur in a subset of observations, then they should specify so, which would lead to additional analyses of fleeing vs no fleeing as a binomial response variable and including only fleeing observation in analyses of MAD as a continuous variable. This is important, as individuals that are approach to a given distance and do not flee are choosing a very different set of trade-offs than individuals approached to the same distance but that stay in place.

L73-79: This paragraph should be written to reduce redundancies.

Field Methods
L82-91: It would be useful to have more detail on the level of protection/permitted activities in protected vs unprotected sites, and whether enforcement/monitoring resources differed between protected areas. The discussion brings up potential differences in fisher compliance; if there was an a priori reason to suspect these differences, it would be good to specify that here.

L94-91: While the length and width of transect are clearly specified, the height is not. Fig 2 shows a cuboid for the transect design, implying that a height and volume are inherent to transects; details/clarification on this should be provided. (Lines 107 refer only to standardized area surveyed, not volume).

L110-113: Here and in Fig 2 there is ambiguity on how the 10-m-length limit of observations related to the 25-m-length of belt transects. Are you dividing the 25-m-length into 2.5 segments? How are you then combining the segments into measures that represent the transects?

L114: define MHI

L134-136: Again, this is an incomplete definition of MAD. See my earlier comments.

Analysis methods
L144-145, If “site” refers to one of two locations (Pūpūkea vs. Hanauma), each including a no-take reserve and an adjacent control area where fishing is permitted, then specify so. Also specify that management refers to protected vs fished (otherwise it is unclear that each “site” includes 2 “management” treatments.)

L148-152 states that “Linear mixed models (LMMs) were used to compare patterns of MAD between sites and management types and assess relationships with fish body length, angle of approach, and water depth. These variables were included as fixed factors in the models, while transect (location) and species were included as random factors. LMMs were developed with the combined data from both sites and for each site separately.”
--First, if what you mean is that site and management type were the main predictors of interest, but other predictors also had to be considered then clarify.
--If the site and management type are the main predictors of interest, why was the above approach taken when it would have been more parsimonious and direct to use a single LMM including management type (protected vs fished) and site (Pūpūkea vs. Hanauma) as fixed effects? Also, form the results tables I see that an interaction between site and management type was included, but this is not clarified in the text.
--It is unclear why transect is used as a random effect. Is this to account for spatial autocorrelation between fish responses recorded at the transect level? Given that transects are nested within sites, this would affect the model’s ability to test for differences between sites (i.e. random effects are somewhat confounded with the fixed effects). An alternative would be to use lat-long matern function that uses the distance between dive sites to account for spatial autocorrelation .

L152-154: It is generally unadvisable to transform the response variable without changing the model structure. (i.e. using glmmPQL instead might be warranted: https://ase.tufts.edu/gsc/gradresources/guidetomixedmodelsinr/mixed%20model%20guide.html

L154-158: It is unclear an ANOVA framework was used to test the relative weight of evidence for various predictors when there are more modern approaches available (i.e. AICc model selection), and why 2 separate sets of GLMMs had to be used. My view is that much more succinct and parsimonious approach would be if a single LMM is used with site, management type, the interaction between the latter two variables, fish body length, angle of approach, and water depth as fixed effects and species as random effects, and then eliminating unnecessary fixed effects via AIC model selection procedures. After fitting the GLM, spatial patterns in the residuals can be examined; if they are significant, then investigate the lat-long matern function.

L165-165: Briefly explain why fourth transformation was required.

L167-168: briefly explain the model simplification process.

L178. Readers are referred to Stamoulis et al. 2018 and to Table 1 for an explanation of habitat variables, but more detail should be given. An appendix with brief definitions of each habitat variable would be helpful.

L185-193. A subset of species distribution models included both management type and MAD. Yet, following the main premise of the paper, MAD and management type are expected to be correlated. The text should clarify how the model selection procedures address this issue.

L215-223. Given my earlier comments on the analysis of MAD as a response variable, I am not confident that these results provide the best statistical inferences from the data.

Discussion
L242-261. Broader context should be given on how MAD response to divers may differ in exploited areas where non-spearfishing methods are used (see my earlier comments).

L264-268. It is good to see reference to Clark 1994 to bring a predation risk context in which larger fish have more reproductive assets to protect and therefore are expected to take less risks. An alternative explanation, which is not mutually exclusive with the former, is that larger fish are targeted for exploitation, and therefore larger fish are more willing to incur fleeing costs than smaller fish.

L269-271: the effects of approach angle are interpreted as “… likely a result of the methodology as opposed to a behavioral response.” While this may be partly true (MAD cannot be smaller than the nearest distance between the fish and the trajectory of the diver), it is also important to state that the predation risk framework predicts a greater MAD when approaches are more direct because a direct approach may convey detection and intent to capture (e.g. Frid and Dill 2002).

L293-295 states that “… management type was not selected as a final predictor for any models meaning it was a comparatively poor predictor of targeted species biomass,” but the interpretation should be more nuanced. It follows from the main premise of the paper that MAD and management type are correlated with each other, and caution is required in interpreting the effects of these variables; it is simplistic to interpret management type (i.e. spatial protection) as a poor predictor of biomass.

L 311-315. Interpretation of the results of species distribution models can be enhanced by using the predation risk framework. As specified in my earlier comments, that framework can guide discussion on the extent to which lower biomass in exploited areas may reflect fishery removals or the combined effects of such removals and the costs of antipredator behaviour, and may also help generate bias correction factors that account for fish behaviors that can be related to characteristics of individual and or species that affect investment in antipredator behavior.

References
Frid, A. & Dill, L. (2002). Human-caused disturbance stimuli as a form of predation risk. Ecol. Soc., 6.
Frid, A., McGreer, M. & Frid, T. (2019). Chasing the light: Positive bias in camera-based surveys of groundfish examined as risk-foraging trade-offs. Biol. Conserv., 231, 133–138.
Heithaus, M.R., Frid, A., Wirsing, A.J. & Worm, B. (2008). Predicting ecological consequences of marine top predator declines. Trends Ecol. Evol., 23.
Samia, D.S.M., Bessa, E., Blumstein, D.T., Nunes, J.A.C.C., Azzurro, E., Morroni, L., et al. (2019). A meta-analysis of fish behavioural reaction to underwater human presence. Fish Fish., 0.
Spitz, D.B., Rowland, M.M., Clark, D.A., Wisdom, M.J., Smith, J.B., Brown, C.L., et al. (2019). Behavioral changes and nutritional consequences to elk (Cervus canadensis) avoiding perceived risk from human hunters. Ecosphere, 10, e02864.

Reviewer 2 ·

Basic reporting

Nicely written.

Introduction is quite clear, but would profit from a more detailed explanation of how SDMs are constructed and used. What predictors are included? How are they combined? What are the outputs? How are they used?

L 63 Need to emphasize that the estimates of biomass differed with survey method, not actual biomass: “…biomass *estimates* of some targeted reef fishes were significantly lower on SCUBA compared to CCR…”

Experimental design

L 82 Study sites: It is important to state clearly whether activities involving divers/snorkelers in the water (such as spearfishing and feeding) are permitted or not in both reserves.

We also need to know the extent to which any prohibitions are enforced within reserves.

Methods: Survey locations are densely distributed within reserves (especially Hanauma Bay), leading to some concerns about pseudoreplication. Please address this possibility, because it could severely restrict the variability in biomass in reserve areas.

Assigning a MAD value of 10m to fishes that were not seen assumes that the fish were present, creating a datum out of no information. Since fishing lowers fish density, isn’t it preferable to simply not record a MAD for fishes that were not seen? To me, this is a major problem, and it also has the possibility of creating an odd variance structure, with a high number of values at the 10m truncation point. How often did this happen?

Validity of the findings

Results:
L 205 Please explain what “targeted by fishers” means. Fishing can include many methods (nets, hook and line) which should not affect fish wariness of divers.

L 209 What is an “open area”? Does it contrast with a closed area?

Discussion:
L 250 – 254 If enforcement at Pūpūkea has been documented as lax, it is not clear why it was included in the binary classification of reserve or no reserve. A better approach that directly addresses the hypothesis would be to estimate the actual fishing pressure on the four areas and use this as a predictor. This would eliminate the need for this speculation.

L 265 “Optimal fitness theory predicts that as reproductive value increases, risk-taking should decrease (Clark 1994).” This is unclear at several levels. First, it is not clear what “optimal fitness theory” is – fitness should be maximized when fitness components are optimized. Second, reproductive value (expected total future reproduction discounted by the probability of being alive) usually is maximum at maturity and declines thereafter, and does not necessarily increase with size. Larger fish are not necessarily more wary because they have higher fecundity. They can detect threats from further away, and they are targeted more heavily than smaller fish.

L 281 “targeted fishes in these marine reserves are likely habituated to non-aggressive human interactions, leading to reduced MAD” This implies that human factors other than fishing affect MAD, yet the introduction and general premise of this paper considers only fishing. Certainly, habituation to humans (or even attraction) has been considered before in the literature, and should be considered here rather than simply mentioned as speculation.

L 293 “In contrast, management type was not selected as a final predictor for any models meaning it was a comparatively poor predictor of targeted species biomass.” True, but this paper points out that management effectiveness differed dramatically between reserve sites, so that a dichotomous “management type” designation is misleading.

L 312 “from the direct effects of fishing pressure, for which MAD provides a proxy.” Again, it is not clear that MAD is a proxy for fishing only.

L 355 – 358 Since this is a multi-authored paper, it is not clear why the Acknowledgements shifts to the use of first person.

Additional comments

This is a useful demonstration of the potential of using a wariness measure (MAD) to improve the predictive performance of Species Distribution Models. However, I have several reservations about methods and assumptions, outlined above, which may improve the paper.

Reviewer 3 ·

Basic reporting

Overall I found this manuscript to be well-written, however, there is some overuse of acronyms throughout. In particular, I would like to see no acronyms in the figures, table captions, and the abstract. These are areas that are communicated to the public, and acronyms are the quickest path to losing your readers' interest. I would suggest that the authors attempt to remove all acronyms throughout their manuscript as well, as we are not worried about word counts these days.

While the authors are correct in their coverage of the literature of diver effects on species distribution models, there is a wealth of research on the effect of diving and spearfishing on the behaviours of fish. Rather than focus their topic on just the specific case of research on diver relationships with species distribution, I would like to see a broader examination of the literature that discussed the impacts of diving and spearfishing on behaviour. Since this is a journal with a broad focus, I believe that would be more appropriate.

In particular, I think there needs to be some discussion of the relationship between size of fish/diver and approach distance. There's good evidence to suggest larger fishes are less affected by divers/spearfishers but small fishes can still display significant behavioural changes to divers, despite not being targeted by spearfishers (ie Bradley, D., Papastamatiou, Y.P. and Caselle, J.E., 2017. No persistent behavioural effects of SCUBA diving on reef sharks. Marine Ecology Progress Series, 567, pp.173-184. vs Benevides, L.J., Cardozo-Ferreira, G.C., Ferreira, C.E.L., Pereira, P.H.C., Pinto, T.K. and Sampaio, C.L.S., 2019. Fear-induced behavioural modifications in damselfishes can be diver-triggered. Journal of Experimental Marine Biology and Ecology, 514, pp.34-40.) and that smaller divers may have less of an impact, as implied by ROV surveys (ie Raoult, V., Tosetto, L., Harvey, C., Nelson, T.M., Reed, J., Parikh, A., Chan, A.J., Smith, T.M. and Williamson, J.E., 2020. Remotely operated vehicles as alternatives to snorkellers for video-based marine research. Journal of Experimental Marine Biology and Ecology, 522, p.151253.), and I think this will require a bit more discussion.

For clarity, I would also suggest the authors change all references of 'fishing pressure' to 'sprearfishing pressure'. The two are fairly distinct and generally one does not associate fishing pressure with impacts on fish behaviour towards divers (I come from a commercial fisheries research background).

I really like the effort the authors went to present their figures so clearly, well done.

Experimental design

Overall, the experimental design and the methods are well explained. One significant aspect I would like to see discussed more is the size ranges of the fishes at each site, which is a confounding factor that, while size has been included in models, it's not clear whether an interaction term between site and size was included.

If spearfishing is having an impact on populations, presumably it is selecting for larger individuals first. Since there is a clear relationship here between size and approach distance, size selection could have affected the patterns more than changes in behaviour. The authors need to present information (ideally simple stats tests and a figure) demonstrating that there is or is not a relationship between mean size and site, and see if including interaction terms between size and site in their model has a significant effect. Otherwise, the claim that fish behaviour is different rather than biomass and size could be misleading.

Since camera can have an effect on accuracy of measurements, I would like to see the model type as well as the lens (focal length included), resolution and framerate used to capture video.

Validity of the findings

In general, the authors are thorough in presenting their results. As indicated above, I would like to see the potential confounding effects of size discussed more thoroughly so that the authors are certain that they are not providing misleading claims regarding behaviour.

Additional comments

The authors present a study documenting fish approach distance to divers in reefs that are protected from spearfishing and reefs that are exposed. The presentation of the figures and results are professional and thorough. In general, my concerns are minor, but some possible confounding factors with fish size between sites need to be examined and tested in their models to ensure their claim that behaviour is changed rather than fish size is valid.

---

## Round 0.2 · Minor Revisions

Thank you for a thorough and careful response to the previous reviews. Two of the three reviewers were available for this version. They both recognize major improvements and have only a small number of additional suggestions.

Reviewer 1 ·

Basic reporting

The revised manuscript is much improved but some issues remain (detailed below).

Experimental design

The revised manuscript has clarified a number of issues about the data collection and analysis. However, as detailed below, I am not convinced that both the GLMM and GLM are needed to analyze drivers of MAD.

Validity of the findings

Except questioning the need for the GLM, I find the findings valid and relevant to conservation.

Additional comments

The authors have done a thorough job addressing reviewer comments and the manuscript is much improved. However, now that other matters have been cleared, I can see that the manuscript is hampered by the conflicting results on the effect of management and angle of approach on MAD, which depend on whether a GLMM or GLM is used (Tables 3 vs 4). The manuscript would be clearer and more succinct by describing the drivers of MAD with the GLMM only (Table 3). The GLMM uses “individual measurements of MAD” while for the GLM “MAD was averaged at the transect level.” According to Table 3, in the GLMM the drivers of MAD are site, fish length, depth, and angle of approach. According to Table 4, in the GLM fish length, depth and management are the drivers (angle drops from the model while management stays). This raises two questions. The first is, which model is a better representation of reality? My view is that the GLMM—which accounts (via random effects) for interspecific variability in MAD and for variable numbers of fish observed per transect—is the better representation. The second question: Why use the GLMM and the GLM if one is more appropriate than the other? In lines 207-208 we are told that MAD must be averaged by transect to be used in the SDM, but that does not resolve the above issues. First, the authors should explain why SDM require that transect-averaged value. If it is because the spatial units used for the SDM are the 25m X 5m dive transects, then this should be specified. Second, it still does not explain why two models with the same predictors were chosen to analyse MAD. If the GLMM is the most appropriate model, then why not use only that one and then simply use the transect-averaged values of MAD for the SDM? Unless the authors have a clear answer to the final question, I strongly recommend dropping the GLM and revising the text accordingly.

Other comments are:
L151-153 states that “Fishes located greater than 10 m in front or 2.5 m to the left or right of the stereo-DOV system as it was moved along the transect were excluded based on minimum visibility encountered and transect dimensions.” It is unclear how “minimum” visibility fits here. If visibility was >10 m during better days but only 10 m during poorer days, and therefore—for consistency—you excluded fish recorded during better days >10 m from the cameras, then clarify the text accordingly.

L237-241. As mentioned in my introductory comments, the spatial units used for the SDMs need to be specified. If they the 25 X 5 m dive transects, then say so.

L275-284 and L300-302. If my advice on using only the GLMM to describe drivers of MAD is followed, then these sections could be rewritten with greater clarity and precision.

L309-“fish drive” requires explanation.

L336: I suggest qualifying the statement as “may have more experience”

L352- the prediction from that literature is for greater flight initiation distance (not MAD). While FID and MAD are related, they are not the same thing and the authors should qualify that sentence accordingly.
L373-5 states that “management status explains variation only between management strata.”
This is very unclear. What is management strata?

Fig 3. It would be better to use the same scales for y axes

Reviewer 2 ·

Basic reporting

Excellent responses to nearly all concerns

Experimental design

No further comment

Validity of the findings

No further comment

Additional comments

In general, the authors have worked very hard to improve this paper. Most of my observations and comments have been adequately addressed. I do have one recommendation to make the paper more palatable to evolutionary ecologists:
L 334 “Optimal fitness theory predicts that as reproductive value increases (Birkeland and Dayton 2005), risk-taking should decrease (Clark 1994).”
As I mentioned before, as far as I can tell there is no such thing as “optimal fitness theory”. Certainly, Clark does not use this phrase, and it does not turn up in any searches I conducted.
As for reproductive value, it can increase with fish size if fecundity increases exponentially with size (as in indeterminant growth) and there is no truncation in survival. This occurs in Clark’s models, and forms the basis for the prediction about risk-taking.
The Birkeland and Dayton paper may be a good reference for dramatic increases in fecundity with size, but it may also contain some misinformation. In the rebuttal the present authors mention “that larger fish produce larvae with higher survivor potential”, and refer to Birkeland and Dayton. This idea remains highly controversial (see, for example, DJ Marshall et al. 2010, The relationship between maternal phenotype and offspring quality: do older mothers really produce the best offspring? Ecology 91:2862-2873). Rather than opening this can of worms, might I suggest a rewording?
“Reproductive value often increases with size in fishes (Birkeland and Dayton 2005), and theory predicts that risk-taking should decrease at higher levels of reproductive value (Clark 1994).”

---

## Round 0.3 · accepted · Accept

Reviewer 1 is satisfied with the changes. Reviewer 2 did not see a need to review the manuscript. Thank you for your careful and thorough responses to the reviewers.

Reviewer 1 ·

Basic reporting

To my satisfaction

Experimental design

To my satisfaction

Validity of the findings

To my satisfaction

Additional comments

I am satisfied with the revisions. Thank you for the hard work!